

# Effect of steady-state aerobic exercise intensity and duration on the relationship between reserves of heart rate and oxygen uptake

Carlo Ferri Marini[1,*], Ario Federici[1,*], James S. Skinner[2], Giovanni Piccoli[1], Vilberto Stocchi[3], Luca Zoffoli[1,4], Luca Correale[5], Stefano Dell'Anna[5,6], Carlo Alberto Naldini[7], Matteo Vandoni[7] and Francesco Lucertini[1]

[1] Department of Biomolecular Sciences –Division of Exercise and Health Sciences, University of Urbino Carlo Bo, Urbino, PU, Italy
[2] Department of Kinesiology, Indiana University, Bloomington, IN, USA
[3] Department of Human Sciences for the Promotion of Quality of Life, University San Raffaele Roma, Rome, RM, Italy
[4] Scientific Research & Innovation Department, Technogym S.p.A., Cesena, FC, Italy
[5] Sports Science Unit, Department of Public Health, Experimental and Forensic Medicine, University of Pavia, Pavia, PV, Italy
[6] Department of Theoretical and Applied Sciences, eCampus University, Novedrate, CO, Italy
[7] Laboratory of Adapted Motor Activity (LAMA), Department of Public Health, Experimental and Forensic Medicine, University of Pavia, Pavia, PV, Italy
* These authors contributed equally to this work.

Corresponding author
Carlo Ferri Marini,
carlo.ferrimarini@uniurb.it

## ABSTRACT

**Background**. The percentages of heart rate (%HRR) or oxygen uptake (%$\dot{V}O_2R$) reserve are used interchangeably for prescribing aerobic exercise intensity due to their assumed 1:1 relationship, although its validity is debated. This study aimed to assess if %HRR and %$\dot{V}O_2R$ show a 1:1 relationship during steady-state exercise (SSE) and if exercise intensity and duration affect their relationship.

**Methods**. Eight physically active males (age $22.6 \pm 1.2$ years) were enrolled. Pre-exercise and maximal HR and $\dot{V}O_2$ were assessed on the first day. In the following 4 days, different SSEs were performed (running) combining the following randomly assigned durations and intensities: 15 min, 45 min, 60% HRR, 80% HRR. Post-exercise maximal HR and $\dot{V}O_2$ were assessed after each SSE. Using pre-exercise and post-exercise maximal values, the average HR and $\dot{V}O_2$ of the last 5 min of each SSE were converted into percentages of the reserves (%RES), which were computed in a 3-way RM-ANOVA ($\alpha = 0.05$) to assess if they were affected by the prescription parameter (HRR or $\dot{V}O_2R$), exercise intensity (60% or 80% HRR), and duration (15 or 45 min).

**Results**. The %RES values were not affected by the prescription parameter ($p = 0.056$) or its interactions with intensity ($p = 0.319$) or duration and intensity ($p = 0.117$), while parameter and duration interaction was significant ($p = 0.009$). %HRRs and %$\dot{V}O_2R$s did not differ in the 15-min SSEs (mean difference [MD] = 0.7 percentage points, $p = 0.717$), whereas %HRR was higher than %$\dot{V}O_2R$ in the 45-min SSEs (MD = 6.7 percentage points, $p = 0.009$).

**Conclusion**. SSE duration affects the %HRR-%$\dot{V}O_2R$ relationship, with %HRRs higher than %$\dot{V}O_2R$s in SSEs of longer duration.

## INTRODUCTION

Aerobic exercise is known to be an effective method for improving health (*Garber et al., 2011*). However, the beneficial effects of aerobic exercise programs vary according to how the components of the FITT-VP principle (*i.e.,* frequency, intensity, type, time, volume, and progression) are manipulated (*American College of Sports Medicine et al., 2021*). Among these components, choosing the proper exercise time (*i.e.,* duration) and intensity is of paramount importance when prescribing aerobic exercise. Indeed, aerobic exercise duration is a key determinant of the cardiorespiratory fitness improvements deriving from a training program (*Lin et al., 2021*). Likewise, aerobic exercise intensity is a crucial consideration in tailoring exercise programs. Selecting the proper intensity, in fact, maximizes the beneficial effects of aerobic training, such as improvements in cardiorespiratory fitness (*Garber et al., 2011*), while minimizing the risks associated with exercise (*American College of Sports Medicine et al., 2021*).

Although moderate, heavy, and severe domains have long been used to divide the aerobic exercise intensity continuum (*Whipp, 1996*) and proved to better discriminate the exercise metabolic stimulus (*Iannetta et al., 2020a*), most aerobic exercise recommendations for the general population (*e.g.,* see *2018 Physical Activity Guidelines Advisory Committee, 2018*) rely on intensity categories (*e.g.,* light, moderate, vigorous). This is probably because using exercise intensity domains to prescribe exercise intensities requires the use of specialized cardiopulmonary equipment or the execution of maximal exercise tests. According to the exercise intensity categories method, aerobic exercise intensity can be prescribed and monitored using oxygen uptake ($\dot{V}O_2$) or heart rate (HR) expressed as percentages of their maximal ($HR_{max}$ and $\dot{V}O_{2max}$, respectively) or reserve (HRR and $\dot{V}O_2R$, respectively) values (*American College of Sports Medicine et al., 2021*). From a theoretical point of view, prescribing the intensity of aerobic exercise using percentages of the reserve values, which are calculated as the difference between maximal and resting values, is more accurate than using percentages of the maximal values because it considers the individual variability in resting HR and $\dot{V}O_2$ and allows correction for nonzero resting values (*Swain, American College of Sports Medicine & of Sports Medicine, 2014*).

The percentages of the two reserves have been found to be highly correlated during incremental exercise, and their relationship is currently considered to be 1:1 (*American College of Sports Medicine et al., 2021*; *Swain, American College of Sports Medicine & of Sports Medicine, 2014*) based on the results of several studies (*Brawner, Keteyian & Ehrman, 2002*; *Byrne & Hills, 2002*; *Lounana et al., 2007*; *Swain & Leutholtz, 1997*), which show that the %HRR-%$\dot{V}O_2R$ relationship is not distinguishable from the identity line (*i.e.,* regression line with slope = 1 and intercept = 0). From a practical standpoint, assuming the 1:1 relationship is appealing because it allows us to apply the same percentage of either HRR or $\dot{V}O_2R$ to obtain the target aerobic exercise intensity. However, the actual association between %$\dot{V}O_2R$ and %HRR during incremental exercise has been questioned in other

studies (*Brawner, Keteyian & Ehrman, 2002*; *Ferri Marini et al., 2021b*; *Hui & Chan, 2006*; *Pinet et al., 2008*; *Swain et al., 1998*), which found that the relationship differed from the identity line.

Notably, regardless of whether there is a 1:1 relationship between %HRR-%$\dot{V}O_2$R, the above-mentioned studies investigated this relationship using only data sampled during incremental exercise tests, while the %HRR-%$\dot{V}O_2$R relationship is currently used to prescribe and monitor prolonged constant-intensity aerobic exercise (*American College of Sports Medicine et al., 2021*). The transferability of the relationships between HR and $\dot{V}O_2$ from incremental to prolonged constant-intensity exercise is a much-debated topic (*Cunha et al., 2011b*; *Wingo, 2015*; *Wingo, Ganio & Cureton, 2012a*). Indeed, although prediction accuracy of prolonged constant-intensity exercise may be increased by accounting for $\dot{V}O_2$ kinetics (*i.e.,* mean response time and $\dot{V}O_2$ slow component) during incremental exercise protocols (*Caen et al., 2020*; *Iannetta et al., 2019*; *Iannetta et al., 2020b*; *Keir et al., 2018*), studies have shown that constant-load exercise intensities are generally not accurately predictable when incremental exercise tests are used to calculate standard percentages of $\dot{V}O_{2max}$, $HR_{max}$, or HRR (*Iannetta et al., 2020a*; *Weatherwax et al., 2019*). Indeed, on one hand, several time-dependent physiological adjustments occur during prolonged exercise, namely cardiovascular drift (*Coyle & Gonzalez-Alonso, 2001*; *Zuccarelli et al., 2018*) and the $\dot{V}O_2$ slow component, inducing increases in HR and $\dot{V}O_2$ over time (*Gaesser & Poole, 1996*; *Whipp & Wasserman, 1972*) which could modify the relationships between HR and $\dot{V}O_2$ found during incremental exercise. In fact, a study by *Cunha et al. (2011b)* which assessed the validity of the %HRR-%$\dot{V}O_2$R relationship during prolonged aerobic exercise at three levels of intensity, showed that the slope of the increase over time was higher in %HRR than in %$\dot{V}O_2$R, resulting in a dissociation between %HRR and %$\dot{V}O_2$R. Moreover, the results showed that %HRR was higher than %$\dot{V}O_2$R (*Cunha et al., 2011b*), which indicates that the relationship between the percentages of the two reserves was not 1:1. On the other hand, Wingo and colleagues (*Wingo, 2015*; *Wingo, Ganio & Cureton, 2012a*), analyzing data deriving from previous studies (*Lafrenz et al., 2008*; *Wingo & Cureton, 2006a*; *Wingo & Cureton, 2006b*; *Wingo et al., 2005*), argued that the dissociation between %HRR and %$\dot{V}O_2$R may be partially mitigated by a decrease in $\dot{V}O_{2max}$ during prolonged aerobic exercise. Therefore, although Wingo and colleagues (*Wingo, 2015*; *Wingo, Ganio & Cureton, 2012a*) did not directly assess whether the %HRR and %$\dot{V}O_2$R had a 1:1 relationship under different exercise conditions, they suggest that studies examining prolonged exercise that did not take into account the increase in %$\dot{V}O_2$R due to the reduced $\dot{V}O_{2max}$, such as the one of Cunha and colleagues (*2011b*), could provide biased %HRR-%$\dot{V}O_2$R relationships.

To date, therefore, the actual association between HRR and $\dot{V}O_2$R during prolonged aerobic exercise is under debate, and how aerobic exercise intensity and duration affect their relationship is largely unknown. Moreover, it is important to point out that although exercise intensity is commonly prescribed using percentages of HR maximal or reserve values (due to the availability, simplicity, and low costs of HR monitors), to date, no study assessed how exercise intensity, duration, and their interaction affect the %HRR-%$\dot{V}O_2$R relationship when HR (expressed as %HRR) is used to prescribe and monitor aerobic

exercise intensity. We hypothesize that, notwithstanding a possible partial mitigation by a decrease in $\dot{V}O_{2max}$, the magnitude of the cardiovascular drift will be greater than the $\dot{V}O_2$ slow component, which will create a dissociation between %HRR and %$\dot{V}O_2$R proportional to the characteristics of the aerobic exercise, namely its intensity and duration. Therefore, this study aimed to assess a) if the percentages of HRR and $\dot{V}O_2$R show a 1:1 relationship and b) the effect of exercise intensity and duration and their interaction on the %HRR-%$\dot{V}O_2$R relationship during prolonged constant-intensity aerobic exercise.

## MATERIALS & METHODS

### Participants

Eight apparently healthy physically active male subjects (age $22.6 \pm 1.2$ years) volunteered to participate in the study.

The inclusion criteria used in the present study were: being males between 18 and 35 years old; having obtained medical clearance to perform maximal exercises; being physically active by performing planned, structured physical activity for at least 30 min at moderate intensity on at least 3 days per week for at least the last 3 months; regardless the type of physical activity performed, having used the treadmill regularly (at least once a week, hence being accustomed to treadmill running), for at least 2 months.

Enrolled participants had at least three years of experience in using the treadmill and had engaged, with a weekly frequency ranging from 3 to 5 sessions per week, in different types of aerobic exercise training aimed at improving either physical fitness or sports performance that involved at least 4 h of aerobic training per week at moderate or higher exercise intensities (of which at least 2 h of vigorous intensity aerobic training per week).

The following exclusion criteria were adopted: use of medications that have been shown to affect cardiorespiratory responses to exercise (enrolled participants did not use any medication during the study); smoking or use of ergogenic substances; recent orthopedic or musculoskeletal injuries that could limit or affect exercise performance.

The study was approved by the Human Research Ethics Committee of the University of Urbino (approval reference number: VN21-10072019) and carried out in accordance with the Declaration of Helsinki.

The subjects were informed of the potential risks and discomfort associated with the testing procedures and provided written informed consent before being enrolled in the study.

### Experimental design

This study employed a randomized cross-over design to assess HR and $\dot{V}O_2$ responses under four different steady-state exercise (SSE) conditions. Participants reported to the laboratory seven times over the course of the study, with at least a three-day interval between each visit. On the first day, the subjects underwent a pre-SSE testing session. Specifically, anthropometric variables and body composition were assessed and pre-exercise and maximal HR and $\dot{V}O_2$ were measured. Two practice trials were then carried out (on separate days) to determine the appropriate treadmill speed to elicit the desired percentages of HRR, which would then be used in the subsequent four SSE testing sessions.

At each of the four SSE testing sessions, participants' maximal HR and $\dot{V}O_2$ were measured during a graded exercise test (GXT) to exhaustion, which was performed immediately after (*i.e.,* with no cessation of exercise or rest) completing the following experimental SSEs (in random order): 15 min at 60% HRR, 15 min at 80% HRR, 45 min at 60% HRR, 45 min at 80% HRR. All assessments were performed at the same time of the day to minimize the possible effect of circadian rhythms on HR and $\dot{V}O_2$ values. See Fig. 1 for a schematic illustration of the experimental design.

The random generation sequences used to select the order of the practice trials and SSE testing sessions were created for each subject using block randomizations performed with the R statistical software (R Core Team, v.3.3 –"Randomizr" package, v.0.12.0). The block dimensions were equal to the number of the conditions to be randomized (*i.e.,* two for the practice trials and four for the SSE testing sessions).

During the SSE testing sessions participants were not allowed to drink and fan airflow was not used because hydrating (*Ganio et al., 2006*) and using fans (*Wingo & Cureton, 2006a*) have been shown to alter cardiorespiratory responses to prolonged aerobic exercise. The testing sessions were performed under controlled room temperature and humidity.

Participants were instructed to avoid changes in their training and dietary habits, to avoid performing vigorous physical activity or consuming alcohol or caffeine both the day before and on the day of each laboratory visit, and to drink plenty of fluids (*Swain, American College of Sports Medicine & of Sports Medicine, 2014*) the day before each testing session. On each testing day, participants arrived at the laboratory following a 3-hour fast and drinking 0.5 L of water 1 h before their scheduled arrival time. Compliance with these instructions was assessed on each testing day using a checklist.

## Assessments and data processing

Exercise tests, practice trials, and SSEs were performed on the Matrix T7xe treadmill (Johnson Health Tech Italia Spa, Ascoli Piceno, Italy) set at 0% grade. Holding onto the side or front bars of the treadmill was not permitted (except, if necessary, for safety reasons). The treadmill grade was set at 0% to a) closely reproduce the validated maximal exercise tests that were used in this study and b) maximize the chances that each participant would be comfortable running at the speeds used during the SSE (using steeper grades would have necessitated slower speeds to elicit the target HR, and those speeds might have been lower than the preferred running speeds).

Participants' $\dot{V}O_2$ and HR were continuously sampled for the entire duration of each practice and testing day (except during anthropometric and body composition assessments). Breath-by-breath $\dot{V}O_2$, carbon dioxide production, and pulmonary ventilation were measured using the K5 portable metabolimeter (COSMED Srl, Rome, Italy), which is a valid and reliable device to measure ventilatory parameters (*Guidetti et al., 2018*). The metabolimeter was calibrated before each test according to the manufacturer's instructions. HR was recorded in beat-to-beat intervals using the Polar V800 HR monitor (Polar Electro Oy, Kempele, Finland).

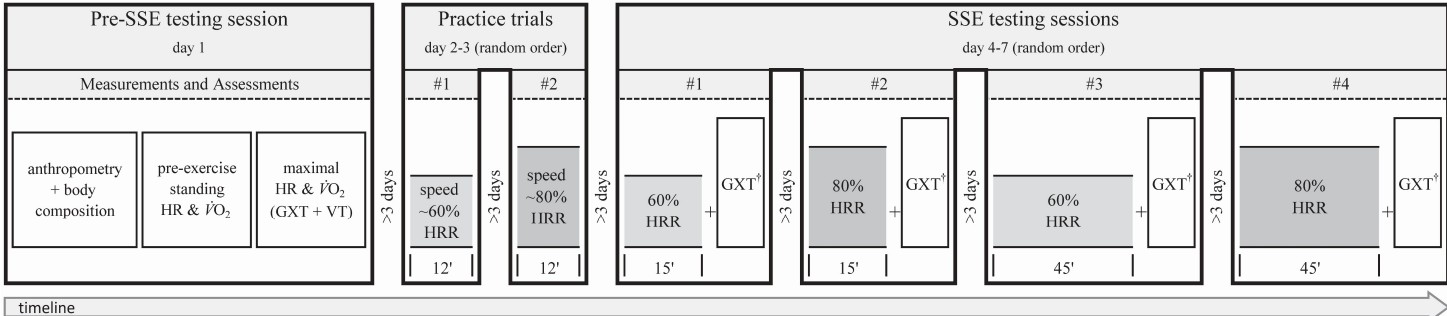

**Figure 1** **Experimental design and timeline of the non-exercise and exercise assessments.** Exercise assessments were always performed by running on a treadmill at 0% grade. SSE, steady-state exercise; HR, heart rate; V̇O₂, oxygen uptake; GXT, graded exercise test; VT, verification trial; HRR, heart rate reserve; †, GXT performed after SSE with no cessation of exercise or warm-up.

### Anthropometry and body composition

Participants underwent the following anthropometric measurements: body mass (barefoot while wearing shorts); height (barefoot and head in the Frankfurt plane); body composition (using bioimpedance analysis - BIA 101, Akern-RJL Systems, Florence, Italy).

### Pre-exercise HR and $\dot{V}O_2$

Participants sat quietly in a chair during the set up of the HR monitor and metabolimeter. HR and $\dot{V}O_2$ were then continuously recorded for 20 min with the subject standing on the treadmill. Resting HR and $\dot{V}O_2$ were measured in the standing position because ACSM's guidelines (*Swain, American College of Sports Medicine & of Sports Medicine, 2014*) recommend measuring the resting values with the subject in a position similar to the one assumed during the prescribed exercise mode. According to ACSM, those values were defined as pre-exercise values (*Swain, American College of Sports Medicine & of Sports Medicine, 2014*).

Both HR and $\dot{V}O_2$ recordings were divided into four 5-min intervals, and the average of each interval was calculated. For each variable, the average of the first 5-min interval was discarded (*American College of Sports Medicine et al., 2021*) and the lowest average value of the three remaining intervals was assumed as the pre-exercise value of standing HR and $\dot{V}O_2$.

### Maximal exercise tests

Participants' $HR_{max}$ and $\dot{V}O_{2max}$ were measured using a personalized GXT running protocol (*da Silva et al., 2012*), which was created for each participant using the Excel spreadsheet provided by *Ferri Marini et al. (2021a)*. Briefly, the spreadsheet allows the automatic creation of incremental exercise protocols consisting of a 3-min warm-up at 40% of the estimated maximal treadmill speed and a personalized ramp protocol that is designed as follows. $\dot{V}O_{2max}$ is estimated by means of the non-exercise model proposed by *Matthews et al. (1999)* The speed yielding the estimated $\dot{V}O_{2max}$ (*i.e.,* the final speed) is then calculated from the estimated $\dot{V}O_{2max}$ according to the ACSM running equation (*American College of Sports Medicine et al., 2021*) and the initial speed of the ramp protocol

is set at 50% of the final speed. The speed increment of each 1-min stage is calculated as the difference between the final and initial speed, divided by 10 min, and multiplied by the number of minutes elapsed from the beginning of the test (warm-up excluded) to the beginning of that given stage. As shown by *da Silva et al. (2012)* using this approach should allow attainment of the maximal speed –and therefore the estimated $\dot{V}O_{2max}$ –approximately at the 10th min of the test.

When the GXT had ended, participants sat quietly for 20 min and then underwent the $\dot{V}O_2$ verification trial (VT) proposed by *Nolan, Beaven & Dalleck (2014)*. Briefly, they performed a 2-min stage at 50% followed by a 1-min stage at 70% of the maximal speed achieved during the GXT. At the beginning of the 4th minute, speed was increased to 105% of the maximal speed achieved during the maximal test and maintained until the subjects could no longer continue. Each participant received strong verbal encouragement to make his maximum effort.

The $\dot{V}O_2$ and HR raw data were smoothed as a 15-breath moving average (*Robergs, Dwyer & Astorino, 2010*) and 5-sec stationary time average (*Midgley et al., 2009*), respectively. For the pre-SSE session, the highest values of $\dot{V}O_2$ and HR recorded during either GXT or VT were assumed to be maximal values if at least one $\dot{V}O_2$ plateau was identified or if the highest HRs recorded during the GXT and VT were within 4 bpm (*Midgley et al., 2009*). GXT and VT $\dot{V}O_2$ plateaus were evaluated separately and, regardless of the test in which the $\dot{V}O_2$ plateau was found, the highest $\dot{V}O_2$ and HR recorded were considered to be the maximal values if the two tests were considered consistent with each other (*i.e.,* if the difference between the highest $\dot{V}O_2$s recorded during the two tests were smaller than the smallest detectable difference). The smallest detectable difference, which reflects the magnitude of change necessary to provide confidence that the change was not resultant of random variation or measurement error, was set at 3.8% (*Guidetti et al., 2018*). In the SSE sessions, since VT was not performed (see Steady-state exercise), the highest values of $\dot{V}O_2$ and HR achieved during the GXT were assumed to be maximal values if a $\dot{V}O_2$ plateau was identified (*Midgley et al., 2009*) or if the highest HR was within 4 bpm from the HR assumed to be maximal in the pre-SSE session. For both the pre-SSE and SSE maximal tests, if a $\dot{V}O_2$ plateau was not found and the highest HR were not within 4 bpm, the entire testing session was repeated after at least 3 days. The $\dot{V}O_2$ plateau was identified as proposed by *Midgley et al. (2009)*. Briefly, breath-by-breath $\dot{V}O_2$ values were expressed as a 30-sec stationary time average. Individual linear regressions (ILR) were performed between the $\dot{V}O_2$ (dependent variable) and work rate (independent variable) recorded during the linear portion of the GXT. The modeled $\dot{V}O_{2max}$ was then calculated as follows: maximal speed achieved during the maximal test x slope of the ILR + intercept of the ILR. The $\dot{V}O_2$ was assumed to have plateaued if the difference between modeled and actual maximal $\dot{V}O_2$ was higher than 50% of the regression slope of the ILR (*Midgley et al., 2009*).

## Practice session trials

Two practice trials were performed on separate days to determine the treadmill speeds needed to elicit 60% and 80% HRR for each participant.

The HRs corresponding to 60% and 80% HRR were calculated with the following formula: ([maximal value − pre-exercise value] x desired percentage) + pre-exercise value. Those HRs were the target HRs to reach during the practice trials. The $\dot{V}O_2$ values corresponding to 40%, 60%, and 80% $\dot{V}O_2R$ were calculated with the same procedure used for HR, and the speeds corresponding to each percentage were calculated (*American College of Sports Medicine et al., 2021*).

The practice trials started with a 3-min warm-up at a speed corresponding to 40% $\dot{V}O_2R$. Then, the exercise intensity was increased to a speed corresponding to either 60% or 80% $\dot{V}O_2R$ (random order). After 3 min, designed to elicit HRs close to the desired fractions of 60% and 80% HRR given the claimed 1:1 relationship between %HRR and %$\dot{V}O_2R$, the speed was adjusted (if needed) every 30 s to reach and maintain the desired target HR. The practice sessions were intended to last no more than 12 min (warm-up included) to avoid possible altered HR responses due to the onset of cardiovascular drift, which generally appears after approximately 15-20 min of aerobic exercise (*Wingo, 2015*). The speeds of the practice trials yielding steady HRs that were closest to the target HRs were used as starting speeds for the SSE conditions.

## Steady-state exercise

SSE started with a 5-min warm-up at the speed (calculated using the ACSM running equation *American College of Sports Medicine et al., 2021*) corresponding to 40% $\dot{V}O_2R$ (calculated using the maximal and pre-exercise values measured during the pre-SSE testing session), followed by either 15 or 45 min of running at either 60% or 80% HRR (random order). After the warm-up, the speed was linearly increased every 30 s to reach the starting speed found during the practice trial in 2.5 min. After about 2 min of running at the starting speed, treadmill belt speed was adjusted to maintain the target HR throughout the SSE session.

To account for possible changes in maximal HR and $\dot{V}O_2$ induced by the SSE, all SSE conditions were immediately followed (*i.e.*, with no cessation of exercise or rest) by the same maximal GXT protocol (warm-up excluded) performed during the pre-SSE testing session (see Fig. 1). The maximal values measured during these GXTs (*i.e.*, post-SSE maximal value) were used to compute the SSEs' %HRR-%$\dot{V}O_2R$ relationships (*Wingo, Ganio & Cureton, 2012a*).

The HR and $\dot{V}O_2$ averages over the last 5 min of each SSE were assumed to represent the specific experimental condition. For each SSE condition, the average $\dot{V}O_2$ and HR values were converted into percentages of HRR and $\dot{V}O_2R$ by calculating the pre-exercise standing HR and $\dot{V}O_2$ values and the post-SSE maximal test results using the following formula: 100 x (SSE average value − pre-exercise value) / (post-SSE maximal value − pre-exercise value).

Rating of perceived exertion (RPE) was also recorded, using the scale 6–20 (*Borg, 1998*), within the last 15-sec of each SSE.

## Statistical analysis

All analyses were performed using SPSS Statistics (IBM, v.20) software, with an $\alpha$ level of 0.05. Participants' compliance with the target HR of each planned SSE intensity was

evaluated using the root mean square error (RMSE), which was calculated as follows: the difference between the actual (5-sec stationary time average) and target HR of each 5-sec interval was calculated, and the sum of the squared differences was divided by the number of intervals before calculating the square root. If the RMSE of a trial was higher than 4 bpm, the experimental SSE session was discarded and repeated. To avoid the confounding effects of the initial HR kinetics, the first 7.5 min of each SSE (*i.e.,* until 5 min after reaching the starting speed found during the practice trial) were excluded from the SSE intensity compliance analyses because they were deemed unrepresentative of the SSE.

Since each repeated measure variable was normally distributed, a three-way (*prescription parameter* × *intensity* × *duration*), factorial repeated measures ANOVA was used to assess if the percentages of the reserve values of the last 5 min of the SSE conditions (dependent variable) were affected by: (a) prescription parameters used to assess exercise intensity (HRR or $\dot{V}O_2R$); (b) intensity of the SSE (60 or 80% HRR); (c) duration of the SSE (15 or 45 min). When a statistically significant result was found, Bonferroni corrected post-hoc pairwise comparison between prescription parameters (HRR *vs.* $\dot{V}O_2R$) were performed.

For each reserve, the changes over time ($\Delta_{45-15}$) were calculated as the difference between the percentage of a given reserve recorded in the 45-min and 15-min SSE conditions. The Cohen's *d* effect sizes (ESs) of the differences between %HRR and %$\dot{V}O_2R$ and of their $\Delta_{45-15}$ were calculated, as proposed by *Cohen (1988)*, by dividing the mean by the SD of the differences between either %HRR and %$\dot{V}O_2R$ or of their $\Delta_{45-15}$, respectively.

## RESULTS

Pre-SSE participant characteristics were (mean ±SD): height, $1.82 \pm 0.06$ m; body mass, $76.3 \pm 8.3$ kg; body mass index, $22.9 \pm 1.8$ kg/m$^2$; body fat percentage, $15.7 \pm 4.9$; pre-exercise HR, $79.6 \pm 10.5$ bpm; $HR_{max}$, $195.3 \pm 11.3$ bpm; pre-exercise $\dot{V}O_2$, $4.6 \pm 0.5$ mL·min$^{-1}$·kg$^{-1}$; $\dot{V}O_{2max}$, $61.5 \pm 8.6$ mL·min$^{-1}$·kg$^{-1}$.

The RMSE between actual and target HR during the SSEs was lower than the pre-imposed cut-off in each SSE (RMSE: mean = 1.96, SD = 0.60, range = 0.93 to 3.25).

The average treadmill speeds needed to maintain constant HRs during the 15- and 45-min SSEs at 60% and 80% HRR are shown in Fig. 2. The average treadmill speeds of the first 5 min (computed after reaching the starting speed) compared to the last 5 min of each SSE decreased of $-4.3 \pm 5.4\%$ (from $10.0 \pm 1.8$ to $9.6 \pm 2.1$ km/h) at 60% and $-5.7 \pm 4.2\%$ (from $14.0 \pm 1.5$ to $13.2 \pm 1.8$ km/h) at 80% HRR during the 15-min SSEs, whereas, during the 45-min SSEs, the treadmill speeds decreased of $-13.2 \pm 6.1\%$ (from $10.2 \pm 1.6$ to $8.8 \pm 1.7$ km/h) at 60% and $-24.1 \pm 6.4\%$ (from $13.8 \pm 1.5$ to $10.5 \pm 1.7$ km/h) 80% HRR.

The HR, $\dot{V}O_2$, and RPE responses to the SSE conditions and the results of the post-SSE GXTs are presented in Table 1.

The percentages of the two *prescription parameters* were not different ($F_{1,7} = 5.216$, $p = 0.056$, partial eta-squared [$\eta_p^2$] = 0.427) and the interactions of *prescription parameters* with *SSE intensity* ($F_{1,7} = 1.150$, $p = 0.319$, $\eta_p^2 = 0.141$) and those of *prescription*

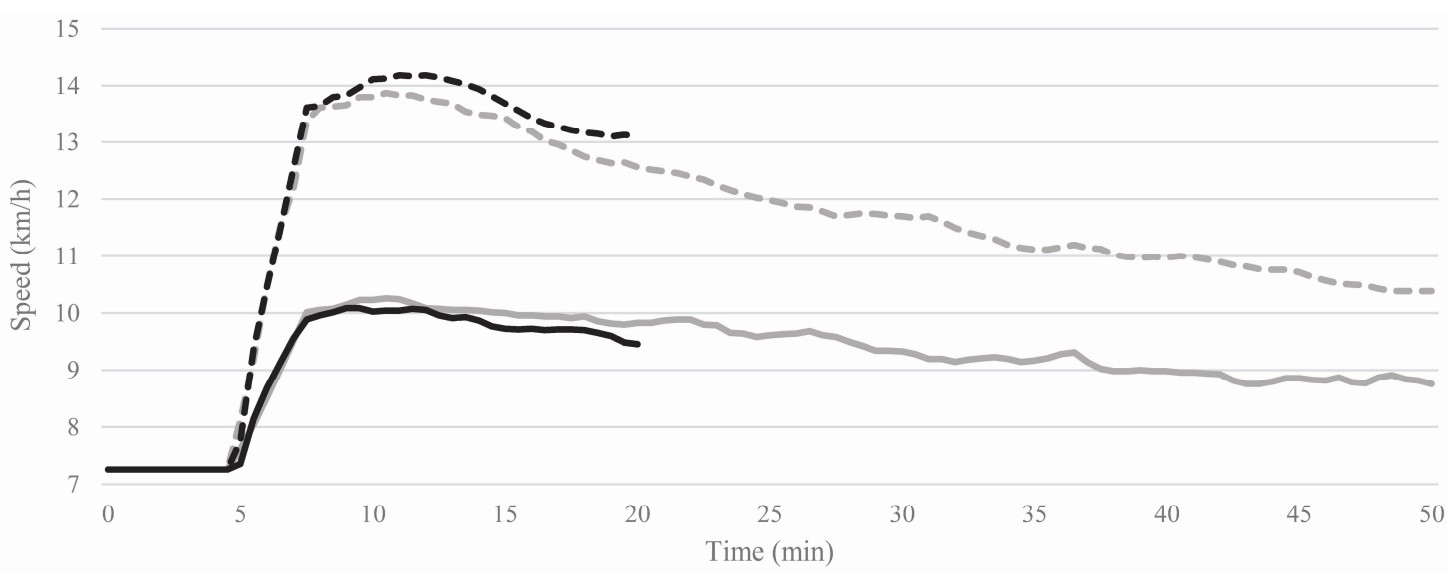

**Figure 2** Average of the treadmill speeds recorded during the 15 (black lines) and 45 (grey lines) min steady-state exercises at 60% (solid lines) and 80% (dashed lines) of heart rate reserve.

**Table 1** Average (ave) HR and $\dot{V}O_2$ in the last 5 min of each SSE, and maximal (max) HR and $\dot{V}O_2$ measured using the graded exercise test to exhaustion performed after each SSE (mean ± SD).

| | SSE at 60% of HRR | | SSE at 80% of HRR | |
|---|---|---|---|---|
| | **15 min** | **45 min** | **15 min** | **45 min** |
| $HR_{ave}$ (bpm) | 150.8 ± 9.7 | 150.1 ± 9.8 | 173.4 ± 10.2 | 173.0 ± 10.5 |
| $\dot{V}O_{2ave}$ (mL min$^{-1}$ kg$^{-1}$) | 39.5 ± 6.3 | 36.4 ± 6.6 | 52.5 ± 5.5 | 45.1 ± 4.5 |
| RPE (6–20) | 10.6 ± 1.2 | 12.3 ± 1.7 | 13.5 ± 1.7 | 14.8 ± 1.7 |
| $HR_{max}$ (bpm) | 195.2 ± 12.9 | 190.9 ± 12.0 | 193.3 ± 10.6 | 194.5 ± 10.4 |
| $\dot{V}O_{2max}$ (mL min$^{-1}$ kg$^{-1}$) | 61.7 ± 6.6 | 58.9 ± 9.4 | 63.0 ± 6.8 | 60.7 ± 8.7 |

**Notes.**

HR, heart rate; $\dot{V}O_2$, oxygen uptake; HRR, heart rate reserve.

RPE (6–20), rating of perceived exertion recorded within the last 15-sec of each SSE using a scale ranging from 6 to 20 points.

*parameters* with *SSE duration* and *SSE intensity* ($F_{1,7} = 3.189$, $p = 0.117$, $\eta_p^2 = 0.313$) were not statistically significant. There was a significant interaction between *prescription parameters* and *SSE duration* ($F_{1,7} = 12.712$, $p = 0.009$, $\eta_p^2 = 0.654$). The post-hoc pairwise comparison analyses showed that the percentages of HRR and $\dot{V}O_2R$ were not different under the 15-min SSE conditions (mean difference [MD] = 0.671 percentage points, $p = 0.717$), whereas the %HRR was higher than %$\dot{V}O_2R$ under the 45-min SSE conditions (MD = 6.722 percentage points, $p = 0.009$).

The differences between %HRR and %$\dot{V}O_2R$ for each SSE, along with the corresponding ESs, are shown in Table 2.

The $\Delta_{45-15}$ showed a decrease in %$\dot{V}O_2R$ from minute 15 to minute 45 for both SSE intensities, while %HRR increased from minute 15 to minute 45 in the SSE at 60% HRR and decreased in the SSE at 80% HRR. The differences in $\Delta_{45-15}$ between %HRR and

**Table 2** HRR and $\dot{V}O_2R$ average percentages in the last 5 min of each SSE condition, their differences (HRR–$\dot{V}O_2R$) and changes over time ($\Delta_{45-15}$).

| | | SSE at 60% of HRR | | | SSE at 80% of HRR | | |
|---|---|---|---|---|---|---|---|
| | | 15 min | 45 min | $\Delta_{45-15}$ | 15 min | 45 min | $\Delta_{45-15}$ |
| HRR (%) | | | | | | | |
| | Mean | 61.7 | 63.4 | 1.6 | 82.6 | 81.4 | −1.3 |
| | SD | 2.8 | 1.5 | 3.2 | 1.6 | 2.3 | 1.5 |
| | ES | | | 0.51 | | | −0.84 |
| $\dot{V}O_2R$ (%) | | | | | | | |
| | Mean | 60.9 | 58.6 | −2.1 | 82.1 | 72.7 | −9.4 |
| | SD | 6.7 | 7.8 | 5.4 | 5.7 | 6.4 | 7.5 |
| | ES | | | −0.39 | | | −1.25 |
| HRR-$\dot{V}O_2R$ (%) | | | | | | | |
| | Mean | 0.9 | 4.8 | 3.8 | 0.5 | 8.6 | 8.1 |
| | SD | 5.3 | 7.0 | 3.7 | 6.3 | 4.8 | 7.3 |
| | ES | 0.17 | 0.69 | 1.00 | 0.07 | 1.81 | 1.12 |

Notes.

HRR, heart rate reserve; $\dot{V}O_2R$, oxygen uptake reserve; SSE, steady-state exercise; $\Delta_{45-15}$, change over time calculated as the difference between the percentage of a given reserve recorded in the 45-min and 15-min SSE conditions; HRR-$\dot{V}O_2R$, difference between the percentages of HRR and $\dot{V}O_2R$ in the 15-min and 45-min SSE conditions and of their $\Delta_{45-15}$; ES, Cohen's *d* effect size.

%$\dot{V}O_2R$ showed a greater decrease in %$\dot{V}O_2R$ than %HRR for both 60% and 80% HRR SSE conditions (see Table 2).

# DISCUSSION

The present study aimed to assess if, during prolonged constant-intensity aerobic exercise, the percentages of HRR and $\dot{V}O_2R$ show a 1:1 relationship and if it is affected by exercise intensity and duration.

The main findings are that the relationship between %HRR and %$\dot{V}O_2R$ is affected by the duration of SSE and that the 1:1 relationship between the percentages of the two reserves is not valid for SSEs of a longer duration. Indeed, the %HRR and %$\dot{V}O_2R$ were not significantly different in the 15-min SSE conditions and showed negligible ESs for both SSE intensities. On the contrary, %HRRs were higher than %$\dot{V}O_2R$s in the 45-min SSEs, respectively with medium and large ESs at 60% and 80% HRR, which points to an effect of the SSE duration on the %HRR-%$\dot{V}O_2R$ relationship. An effect of SSE duration on the %HRR-%$\dot{V}O_2R$ relationship is also supported by their changes over time, which were greater (*i.e.,* tending toward higher increases or smaller decreases over time) for %HRR than for %$\dot{V}O_2R$ in both SSE intensities. This suggests a dissociation between the percentages of the two reserves over time, which induced higher %HRR than %$\dot{V}O_2R$ for the 45-min SSE conditions.

This study cannot be easily compared with the available literature due to the differences between exercise protocols and aims of the studies. The results of a few studies (*Wingo, 2015*; *Wingo & Cureton, 2006b*; *Wingo, Ganio & Cureton, 2012a*; *Wingo et al., 2005*), however, may be partially compared to those of the present investigation. In their reviews (*Wingo,*

*2015*; *Wingo, Ganio & Cureton, 2012a*), Wingo et al. analyzed published and unpublished data from their previous studies (*Lafrenz et al., 2008*; *Wingo & Cureton, 2006a*; *Wingo & Cureton, 2006b*; *Wingo et al., 2005*) and concluded that changes over time of the %HRR-%$\dot{V}O_2$R relationship were associated and that their relationship was described by the following equation: % $\Delta_{45-15}\dot{V}O_2R = 0.83 \times\% \Delta_{45-15}HRR + 1.98$. Unfortunately, the equation was not compared to the identity line (*i.e.,* slope = 1 and intercept = 0). Hence, these results do not allow us to assume that the %HRR-%$\dot{V}O_2$R relationship is maintained after prolonged aerobic exercise. Moreover, although only two (*Wingo & Cureton, 2006b*; *Wingo et al., 2005*) of the four studies examined in Wingo's reviews (*Wingo, 2015*; *Wingo, Ganio & Cureton, 2012a*) reported the actual %HRR and %$\dot{V}O_2$R after prolonged aerobic exercise, the %HRR was higher than %$\dot{V}O_2$R in every condition and study, with the following MD: 8.5% after 15 min and 4.9% after 45 min of aerobic exercise at a constant power output (*Wingo et al., 2005*; 9.5% after 15 min and 9.9% after 45 min of aerobic exercise at a constant power output (*Wingo & Cureton, 2006b*); 6.7% after 15 min and 16.9% after 45 min of aerobic exercise at a constant HR (*Wingo & Cureton, 2006b*). Therefore, in line with our results, these findings seem to disprove the maintenance of a 1:1 relationship (*i.e.,* equality) between the percentages of HRR and $\dot{V}O_2$R during prolonged aerobic exercise; this should not be inferred from the results in Wingo's reviews (*Wingo, 2015*; *Wingo, Ganio & Cureton, 2012a*) or articles (*Wingo & Cureton, 2006b*; *Wingo et al., 2005*; *Wingo, Stone & Ng, 2020*). On the other hand, the investigation by *Cunha et al. (2011b)* did aim to assess if the %HRR-%$\dot{V}O_2$R relationship was 1:1 during prolonged aerobic exercise of different intensities. They found that the %HRR was higher than %$\dot{V}O_2$R, hence the %HRR-%$\dot{V}O_2$R relationship was not 1:1, with a MD of 7.2%. The authors also reported no significant differences between the increases in the percentages of the two reserve values over time but claimed that these differences tended to be higher at higher exercise intensities. However, the authors (*Cunha et al., 2011b*) did not measure maximal HR and $\dot{V}O_2$ following the prolonged aerobic exercise conditions. Therefore, the %HRR-%$\dot{V}O_2$R relationship could have been biased due to changes in maximal HR and $\dot{V}O_2$ induced by aerobic exercise (*Wingo, Ganio & Cureton, 2012a*).

Contrary to our initial hypothesis, the interaction between the duration and intensity of the SSEs did not significantly affect the %HRR-%$\dot{V}O_2$R relationship. However, the lack of a statistically significant interaction could be due to the small sample size of this study rather than the lack of an actual interaction. Indeed, when the ES are interpreted, in the 45-min SSE condition, the ES of the differences between %HRR and %$\dot{V}O_2$R at an intensity of 80% HRR was >2.5 times greater than that found at 60% HRR. On the other hand, the ESs were similar between the two intensities in the 15-min SSE, suggesting that SSE intensity may have an effect solely in longer SSE conditions when SSE intensity could interact with SSE duration. Indeed, the effect sizes for the ANOVA independent variables (*i.e.,* $\eta_p{}^2$) of the *prescription parameters* and *SSE intensity* alone are considerably lower than those of the interaction among *prescription parameters*, *SSE intensity*, and *SSE duration*. The differences between the changes over time of %HRR and %$\dot{V}O_2$R, in our opinion, also support the possible interaction effect between SSE duration and intensity. In fact, in the lower intensity SSE conditions, the difference between $\Delta_{45-15}$ in %HRR were 3.8%

higher than $\Delta_{45-15}$ in %$\dot{V}O_2R$, whereas in the higher intensity SSE conditions, the $\Delta_{45-15}$ in %HRR were 8.1% higher than $\Delta_{45-15}$ in %$\dot{V}O_2R$. Thus, on average, there was a higher dissociation over time between the two reserves when the SSE intensity was higher.

Among the aforementioned studies (*Cunha et al., 2011b*; *Lafrenz et al., 2008*; *Wingo & Cureton, 2006a*; *Wingo & Cureton, 2006b*; *Wingo et al., 2005*), only one (*Wingo & Cureton, 2006b*) employed prolonged aerobic exercise bouts performed at a constant HR, but the participants' compliance to the target SSE intensities was not assessed. Therefore, one strength of the present investigation is that participants' compliance with the target intensity was assessed by calculating the RMSE between the actual and target HR for each subject and SSE condition. Indeed, during prolonged aerobic exercise at constant power output, physiological adjustments occur, namely cardiovascular drift and the $\dot{V}O_2$ slow component, which have been shown to increase in relation to exercise intensity (*Cunha et al., 2011b*). In the present study, the relative exercise intensity was held constant throughout the SSE of each condition by maintaining a constant HR because, unlike $\dot{V}O_{2max}$ (*Wingo, Ganio & Cureton, 2012a*), $HR_{max}$ has been widely shown not to be altered by prolonged aerobic exercise (*Ganio et al., 2006*; *Lafrenz et al., 2008*; *Wingo & Cureton, 2006a*; *Wingo & Cureton, 2006b*; *Wingo et al., 2012b*; *Wingo, Stone & Ng, 2020*). Therefore, using fixed percentages of $\dot{V}O_{2max}$ or $\dot{V}O_2R$ or speeds corresponding to certain $\dot{V}O_2$ would have yielded different relative intensities throughout the SSEs. Indeed, as expected, after reaching the target HRs, the treadmill speeds had to be continuously decreased to maintain a constant HR during each SSE. Although the changes in the treadmill speeds showed similar trends during the first 15 min of the 15- and 45-min SSEs performed at the same exercise intensity, the treadmill speeds seemed to decrease more during the SSEs at 80% HRR compared to those at 60% HRR. Moreover, the average percent decrease from the first to the last 5 min of each SSE seemed to be higher in SSEs of longer durations and higher intensities, supporting a possible effect of the interaction between exercise intensity and duration on the cardiovascular drift.

The RPEs were on average higher during SSE having higher intensities and longer durations. Although higher RPEs values at higher intensities was to be expected, the tendency of reporting higher RPE during SSE of longer durations seems to point out a possible dissociation between physiological parameters, which stay constant (HR) or decline over time ($\dot{V}O_2$), and the psychological perceived exertion, which, on the other hand, increase over time.

A strength of the present study is that pre-exercise HR and $\dot{V}O_2$ were directly measured. Indeed, although pre-exercise $\dot{V}O_2$ can be assumed to be 3.5 mL $\cdot$min$^{-1}$ $\cdot$kg$^{-1}$ (*American College of Sports Medicine et al., 2021*), this estimation might not be appropriate for each individual (*Byrne et al., 2005*). In this regard, a limitation of the present study is that pre-exercise HR and $\dot{V}O_2$ were not measured during the SSE sessions; hence, variations in the pre-exercise values could have affected the calculation of the reserve values. However, due the standardization procedures adopted and the verification of the pre-exercise instructions, this bias should have been minor.

A limitation of the present study is the homogeneity of the participants, who were all involved in moderate and vigorous intensity aerobic training and accustomed to train

at intensities close to those performed during the SSEs. Indeed, factors such as running economy (which is associated with the prior specific exercise experience) and biomechanical differences at different exercise intensities (which, in this study, are relevant both between and within SSEs, considering a mean reduction of 3.3 km/h during the 45-min SSE at 80% of HRR), could be an important confounding factor to consider when assessing HR-$\dot{V}O_2$ relations. Therefore, the results of this study should be carefully extrapolated to other exercise intensities, modalities, and populations.

Another limitation of the present study is that the warm-up speed was estimated using the ACSM running equation to reduce the testing burden (*i.e.,* to avoid performing an additional practice trial to find the treadmill speed corresponding to 40% HRR). However, we believe that this is a minor inaccuracy, with limited effects on the results of the present study, because (a) the relatively low intensity of the warm-up (*i.e.,* about 40% $\dot{V}O_2R$), (b) the relatively short period of the warm-up (first 5-min), and (c) each individual used the same (individualized) exercise intensity for all SSEs.

In the present study, the site-specific technical error (a combination of the biological variability and measurement error of the metabolic cart (*Weatherwax et al., 2018*)) of the dependent variables was not assessed, which is a limitation. However, considering that %HRRs were kept constant during SSEs and assuming a possible technical error of five percentage points in %$\dot{V}O_2R$, individual responses seem to support the effect of SSE duration. Indeed, during the 15-min SSEs, %HRRs were more than five percentage points different from %$\dot{V}O_2R$ in three participants at 60% of HRR (one higher and two lower) and in four participants at 80% of HRR (two higher and two lower), while, during the 45-min SSEs, %HRR was five percentage points higher than %$\dot{V}O_2R$ in three (at 60% of HRR) and six (at 80% of HRR) participants and lower in zero participants. Therefore, during SSEs of longer duration, the number of participants having %HRR five percentage points higher than %$\dot{V}O_2R$ increases, while the number of participants having it lower decreases. This trend seems to be more evident during SSEs of longer duration at higher intensity since most of the participants' %HRR were five percentage points higher than %$\dot{V}O_2R$, which supports a possible interaction effect between SSE duration and intensity.

**Future perspective**

Although the existence of a 1:1 relationship between %HRR and %$\dot{V}O_2R$ is accepted by some (*e.g.,* see *American College of Sports Medicine et al., 2021*), its validity is under debate during both incremental (*Cunha, Farinatti & Midgley, 2011a*; *Ferri Marini et al., 2021b*) and prolonged (*Cunha et al., 2011b*; *Wingo, 2015*; *Wingo, Ganio & Cureton, 2012a*) aerobic exercise. In addition, it is worth noting that rather than using the widely accepted intensity *categories* (*American College of Sports Medicine et al., 2021*) relative to HRR or $\dot{V}O_2R$ percentages (*e.g.,* light, moderate, vigorous, etc.), exercise intensity can also be prescribed based upon a different approach that considers exercise intensity *domains* (*e.g.,* moderate, heavy, severe) identified by physiological demarcation points, such as ventilatory or lactate thresholds and maximal lactate steady-state or critical power (*Whipp, 1996*). Importantly, it has been shown that there is poor agreement between exercise intensities *categories* and *domains* (*Iannetta et al., 2020a*) and that using an approach based on physiological

demarcation points positively affects the number of responders (*i.e.,* participants whose $\dot{V}O_{2max}$ increased) after an aerobic training program (*Weatherwax et al., 2019*). Therefore, a limitation of the present study is that the exercise intensity *domains*, and their effect on the HR, $\dot{V}O_2$, PO, and RPE relationships, were not considered. Since tailoring the aerobic exercise stimulus is crucial to safely obtaining health benefits, further studies are needed to investigate, and possibly model, the true nature of the %HRR-%$\dot{V}O_2$R relationship during prolonged aerobic exercise and its relations with other demarcation points. Hopefully, those studies will examine if and how exercise intensity and duration and several other factors, such as environmental conditions, hydration status and fluid replacement (etc.), affect the HR-$\dot{V}O_2$ relationships during prolonged aerobic exercise.

### Practical applications

The present study suggests that the assumption that %HRR and %$\dot{V}O_2$R are equal in prolonged aerobic exercise could lead to over- and under-estimation errors of the actual exercise $\dot{V}O_2$ and HR when a target HR or power output is used as an intensity prescription parameter.

The present investigation thus brings to light the following important considerations, which exercise professionals should be aware of when prescribing exercise intensity based on a 1:1 relationship between %HRR and %$\dot{V}O_2$R. When HR is used, the exercise $\dot{V}O_2$, and thus intensity and caloric expenditure, could be lower than expected. When constant power output (*e.g.,* treadmill grade and speed) is used, the exercise HR could be higher than expected.

From a practical standpoint, although the error of assuming a 1:1 relationship between %HRR and %$\dot{V}O_2$R may be considered trivial in the 15-min SSE conditions (less than 1% on average), it becomes meaningful in the 45-min SSEs, with MD of 4.8% and 8.6% at 60% and 80% HRR, respectively. Moreover, the individual variability of the differences between %HRR and %$\dot{V}O_2$R in the SSEs was at least 4.8% (SD). These estimation errors, due to the MD between %HRR and %$\dot{V}O_2$R and their individual variability, in certain individuals, could yield exercise intensity outside the prescribed ranges during prolonged exercises.

## CONCLUSIONS

The results of the present investigation show an effect of SSE duration on the %HRR-%$\dot{V}O_2$R relationship, with %HRRs found to be greater than %$\dot{V}O_2$R in SSEs of longer duration. In addition, it appears that the effect of SSE duration on the %HRR-%$\dot{V}O_2$R relationship could be partially affected by SSE intensity, with a larger dissociation between %HRR and %$\dot{V}O_2$R at higher exercise intensities. The present study thus highlights that using the HR-$\dot{V}O_2$ relationships derived from incremental exercise may be inadequate to prescribe prolonged constant-intensity aerobic exercise.

### Funding

The authors received no funding for this work.

## Competing Interests

Matteo Vandoni is an Academic Editor for PeerJ. Luca Zoffoli is employed by Technogym S.p.A., Cesena, FC, Italy.

## Author Contributions

- Carlo Ferri Marini conceived and designed the experiments, performed the experiments, analyzed the data, prepared figures and/or tables, authored or reviewed drafts of the paper, and approved the final draft.
- Ario Federici, James S. Skinner, Giovanni Piccoli and Vilberto Stocchi conceived and designed the experiments, authored or reviewed drafts of the paper, and approved the final draft.
- Luca Zoffoli analyzed the data, authored or reviewed drafts of the paper, and approved the final draft.
- Luca Correale performed the experiments, analyzed the data, authored or reviewed drafts of the paper, and approved the final draft.
- Stefano Dell'Anna and Carlo Alberto Naldini performed the experiments, analyzed the data, prepared figures and/or tables, and approved the final draft.
- Matteo Vandoni conceived and designed the experiments, performed the experiments, prepared figures and/or tables, authored or reviewed drafts of the paper, and approved the final draft.
- Francesco Lucertini conceived and designed the experiments, prepared figures and/or tables, authored or reviewed drafts of the paper, and approved the final draft.

## Ethics

The following information was supplied relating to ethical approvals (i.e., approving body and any reference numbers):

Human Research Ethics Committee of the University of Urbino (approval reference number: VN21-10072019).

## Data Availability

The raw measurements are available in the Supplementary File.

## Supplemental Information

Supplemental information for this article can be found online at http://dx.doi.org/10.7717/peerj.13190#supplemental-information.

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
