# Peer review of "Effect of steady-state aerobic exercise intensity and duration on the relationship between reserves of heart rate and oxygen uptake"

_PeerJ, doi:10.7717/peerj.13190_

## Round 0.1 · original submission · Minor Revisions

Thank you for your manuscript. Please pay attention to each of the reviewers' comments and feel free to ask me for any clarification. I look forward to your revision. Scotty

Reviewer 1 ·

Basic reporting

The author conducted a well structured and organized study with professional language used to report findings. The tables and figures are appropriate for the information being discussed. The hypothesis and research aims were clearly stated. The only suggested reference additions to further highlight the importance of what the authors accomplished relates to the following:

Please consider changing the ACSM guidelines reference from the 10th ed to the 11th ed in order to highlight the current recommendations (does not impact the writing or interpretations of the study).

Line 80-83: This information is critical insight into the exploration of HRR and VO2R relationship. It is suggestion that more background be provided identifying the use of VO2R in previous research and within the ACSM guidelines that encourages the use of VO2max – 3.5 ml/kg/min for VO2R calculation. The resting VO2 value of 3.5 is acting as a baseline/zeroed value which contradicts this important line in the introduction and reemphasizes the importance of your research methodology to collect each participant’s resting HR and VO2. I would suggest including the following article and have not seen much, if anything in the literature addressing this concern with the use of VO2R and metabolic equivalents – Byrne et al (2005) Metabolic equivalent: one size does not fit all J Appl Physiol 99: 1112-1119

Experimental design

The methods and aims of the study were clearly identified. However, there are some suggestions/considerations to better understand the methodology:

Line 148: Was all aerobic activity included or were participants recruited based on running experience with treadmill use?

Line 150: Please clarify what was considered to be ‘2 months of treadmill experience.’

Line 171-172: Please clarify the measurement of maximal HR and VO2 – was this an assessment of true maximal values or the highest achieved submaximal values during steady state? Was a maximal test performed immediately following steady state exercise?

Line 245-267: Please elaborate and clarify on the testing methodology used, especially with the addition of the verification protocol, which was great to see. There appears to be discrepancies in identification in maximal values compared to what has been previously established in the literature. For example, the following statement “For the pre-SSE session, the highest values of V̇O2 and HR recorded during either GXT or VT were assumed to be maximal values if at least one V̇O2 plateau was identified or if the highest HRs recorded during the GXT and VT were within 4 bpm (Midgley et al. 2009)” suggests that a plateau noted in either the GXT or VT would identify a VO2max – How was it handled if a plateau was noted in the GXT, but maximal VO2 values in the VT were higher? What was the error measurement between GXT and VT trial that was acceptable? If plateaus in VO2 in the GXT and VT bouts, how closely did these values need to be to each other in order to confirm a maximal bout was achieved?

Line 255-258: How was the GXT performed after the SSE session? Did participants cool down, recover and then perform the GXT or were they taken from their steady state level to maximal with incremental speed increases? This information is not clear within the methods section and more information/insight is needed in order to truly grasp the significance of the findings.

Line 259-260: If a participant did not achieve ‘max’ following the SSE trial within the GXT and needed to repeat, was the entire SSE session repeated or just the GXT?

Line 292-294: Please provide rationale for the use of VO2R with the ACSM metabolic calculation for running but then used HRR for the exercise protocols. Wouldn’t the use of the VO2R from a standardized, one-size fit all equation encourage the use of different relative warm-up intensities which could impact the results of the exercise trial?

Line 312: RMSE was a great addition to the study!

Methods (general): Were ratings of perceived exertion not gathered? If they were, please report them since they would provide considerable insight into psychological considerations that may influence HR response rather than purely being a physiological response.

Methods (general): The authors are commended for obtaining resting VO2 values to use in assessing VO2R and for also completing GXTs following each SSE trial. However, if it is noted that changes in the exercise session may influence maximal values, would it not also then be important to identify that resting VO2 values on the days of the SSE trails may also vary which would alter the percentages of VO2R? This is something that should be noted, at the very least, as a limitation.

Validity of the findings

Please include the absolute values of speed during the testing to compliment the decrease in percentages.

With the inclusion criteria being 30 min of moderate, structured PA on at least 3 days a week, couldn’t this also impact the result found? If participants were not aerobically active beyond this 30 min for 3 days a week or did not maintain aerobic intensities above mod (80% HRR/VO2R being categorized toward the higher end of vigorous activity, wouldn’t this decrease is relationship at higher intensities for a longer duration be expected based on running economy and mechanical movement considerations?
Line 385-389: Possibly. However, with reductions in running speed averages of nearly 4 kmh, there will be considerable changes in mechanical differences due to speed alone which would impact tissue level O2 demand, especially in the upper extremity which would drive VO2 down rather than maintain. The findings of the current study may be due to familiarity of intensity level based on pre-trial training alone.

Perhaps the largest consideration that has not been addressed in the present study discussion is the evaluation that HRR and VO2R may not be a 1:1 relationship with prolonged steady state… but this assumption has been made on group average data rather than individually driven interpretations. It has been well established that physiological thresholds (VT/VT1, VT2/RCP, lactate) occur at a wide range of HR intensity values. Therefore, the longer duration 1:1 HRR to VO2R considerations may be an inappropriate group average assessment and may underpin the reason for needing to decrease treadmill speeds throughout the higher intensity, longer duration sse trial. The above is an incredibly important consideration that should be noted within the paper as the field is (slowly) identifying fundamental issues and concerns with standardize group approaches to the percentages of HRmax, HRR, VO2max, and VO2R to be used at the individual level. It should also be noted that measurement error of the metabolic system used was not considered, as well. No metabolic analyzer has a perfect measure, and the error may impact the results, especially due to the low population size. Not a flaw of the study, but a suggestion to consider mentioning.

The authors are strongly encouraged to include a limitations section to help strengthen their paper by acknowledging some of the larger considerations and to highlight changes/alterations in future research methodologies related to the topic.

The authors are urged to use caution with the language within the practical applications section. Specifically, “When HR is used, the exercise V̇O2, and thus intensity and caloric expenditure, could be lower than expected; this implies a less effective exercise stimulus.” Furthermore, including the usage of bike wattage in the subsequent sentence is an inappropriate application of the findings since HR and VO2 responses during leg cycling do not align with those seen during running/walking on a treadmill.

Additional comments

No additional comments

·

Basic reporting

Well written - no English concerns.
No grammar or other punctuation edits needed.

Experimental design

Robust design with gold standard and well-defined procedures.

Validity of the findings

Data is robust and authors are to be commended.
My main comments are three-fold:
1. What is the expected biologically variability in the measures obtained (i.e., %HRR and %VO2R)? In our experience, it can at least be 5%. As such, how does this potentially impact the interpretation of your findings?
2. What are the individual responses? Prescribing exercise is done for the individual and not group. Were some individuals more precise on the %HRR/%VO2R relationship, especially with longer durations? And vice versa, was there even greater disparity between the relationship in other participants?
3. Near the end of the discussion, the authors make the following comment: "unexpected modification in the HRR-V̇O2 relationships could have possible drawbacks, which can range from a less effective exercise stimulus to a higher risk of exercise-related events." My question is do we really have evidence to show that the difference in the HRR/VO2R relationship will actually contributed to these scenarios? I would argue no and unless the authors can demonstrate otherwise, I think this statement is too much conjecture.

Additional comments

Overall the authors are to be commended. The study is well-designed and executed. Extremely well-written. I respectfully ask the authors to consider my remarks in the previous section.

---

## Round 0.2 · accepted · Accept

Thank you for your attention to the comments of the reviewers and the thoroughness of your responses. Congratulations on an excellent paper!